# The Satellite DNA Catalogues of Two Serrasalmidae (Teleostei, Characiformes): Conservation of General satDNA Features over 30 Million Years

**DOI:** 10.3390/genes14010091

**Published:** 2022-12-28

**Authors:** Caio Augusto Gomes Goes, Natalia dos Santos, Pedro Henrique de Mira Rodrigues, José Henrique Forte Stornioli, Amanda Bueno da Silva, Rodrigo Zeni dos Santos, Jhon Alex Dziechciarz Vidal, Duílio Mazzoni Zerbinato de Andrade Silva, Roberto Ferreira Artoni, Fausto Foresti, Diogo Teruo Hashimoto, Fábio Porto-Foresti, Ricardo Utsunomia

**Affiliations:** 1Faculty of Sciences, São Paulo State University (UNESP), Bauru 17033-360, SP, Brazil; 2Institute of Biological Sciences and Health, Federal Rural University of Rio de Janeiro, Seropédica 23890-000, RJ, Brazil; 3Department of Structural, Molecular and Genetic Biology, State University of Ponta Grossa, Ponta Grossa 84030-900, PR, Brazil; 4Department of Genetics and Evolution, Federal University of São Carlos, São Carlos 13565-905, SP, Brazil; 5Department of Structural and Functional Biology, Institute of Biosciences, São Paulo State University, Botucatu 18618-970, SP, Brazil; 6Aquaculture Center of UNESP, São Paulo State University, Jaboticabal 14884-900, SP, Brazil

**Keywords:** cytogenomics, satellite DNA, fish cytogenetics, repetitive DNA

## Abstract

Satellite DNAs (satDNAs) are tandemly repeated sequences that are usually located on the heterochromatin, and the entire collection of satDNAs within a genome is called satellitome. Primarily, these sequences are not under selective pressure and evolve by concerted evolution, resulting in elevated rates of divergence between the satDNA profiles of reproductive isolated species/populations. Here, we characterized two additional satellitomes of Characiformes fish (*Colossoma macropomum* and *Piaractus mesopotamicus*) that diverged approximately 30 million years ago, while still retaining conserved karyotype features. The results we obtained indicated that several satDNAs (50% of satellite sequences in *P. mesopotamicus* and 43% in *C. macropomum*) show levels of conservation between the analyzed species, in the nucleotide and chromosomal levels. We propose that long-life cycles and few genomic changes could slow down rates of satDNA differentiation.

## 1. Introduction

Satellite DNAs (satDNAs) comprise a group of highly heterogenous sequences that constitute a substantial fraction of nuclear DNA, being the major DNA component of heterochromatin and occupying preferentially pericentromeric and/or subtelomeric regions [1,2]. In general, these sequences are organized as long arrays of monomers repeated in tandem that can span up to dozens of megabases [3,4,5]. While highly abundant in eukaryotic genomes and often associated with important chromosomal structures, such as centromeres and telomeres, we still do not fully understand the role of satDNAs in the cell, although particular cases of satDNAs acting as noncoding RNAs [6,7] in the assembly of centromeric chromatin and heterochromatin stabilization were already reported [8].

Once a satDNA emerges and is established in a species/population, its monomer sequences evolve in concert, in which polymorphisms can be homogenized and fixed in a reproductive group [2]. This process would inexorably result in a quantitative and qualitative differentiation of satDNA profiles in reproductive isolated groups [9,10]. However, considering a broader evolutionary scenario, other factors could also affect satDNA evolution, including functional constraints, mutation rates and generation times, among others [10,11,12,13,14]. In this context, the first step towards a better comprehension about satDNA biology is the massive characterization of catalogues of satDNAs, the so-called satellitomes [15], in related species, which was unreachable until few years ago due to the absence of adequate tools [16].

In the last few years, the first satellitomes from distinct organisms were characterized, including plants and animals [5,10,15,17,18,19,20], and this helped to elucidate and confirm the genomic organization and the main aspects of short- and long-term evolution of satDNAs. Particularly, in fishes, multiple satellitomes from species within the Characiformes order were prospected. Each of the mentioned studies was carried out for specific aims, such as deciphering the clustering of satDNAs in sex and/or supernumerary chromosomes, as well as detect sharing of satDNAs. Considering 22 families within Characiformes [21], catalogues of satDNAs are available for species of only 5 families (Crenuchidae, Anostomidae, Prochilodontidae, Triportheidae and Characidae) [18,22,23,24,25,26,27].

Serrasalmidae is a characiform family that comprises the pacus and piranhas, currently composed by 16 genus and approximately 101 valid species [28] that are distributed in the Orinoco, Amazon and Paraná–Paraguay River basins [29,30]. This family can be divided into three well-supported subfamilies [31], with a substantial karyotype conservation: Myleinae, with 2*n* = 58 [32]; Serrasalminae, with 2*n* = 58–64 [33,34,35]; and Colossomatinae, 2*n* = 54 chromosomes [34,36,37]. This latter group harbors the widely produced genera *Piaractus* and *Colossoma*, the major native fish of aquaculture industry in South America [38]. Remarkably, interspecific hybrids (“tambacu” and “paqui”) are frequently produced in the fish farms [39], in spite of the long divergence time between both species [31]. Such a possibility could be associated to the low karyotype changes during the evolutionary history of both genera.

Considering that few apparent genomic changes occurred along Serrasalmidae history, questions regarding the evolution of satDNA (one of the fastest evolving DNA sequences) in these taxa emerged. Here, we combined cytogenetics, molecular genetics and genomics to characterize and compare the satellitomes of *P. mesopotamicus* and *C. macropomum*, which diverged approximately 30 mya [31]. We found that both species share a significant number of satDNAs with at least 80% of similarity (*n* = 15, representing 50% and 32% of satDNAs from *P. mesopotamicus* and *C. macropomum*, respectively). In addition, cytogenetic analyses revealed that almost all the shared satDNAs remained conserved in terms of chromosomal position, with slight changes in number of clusters. The general karyotype conservation between both species is accompanied by the long-term conservation of satDNA sequences and could be associated with low genetic differentiation in this group.

## 2. Materials and Methods

### 2.1. Sampling, Cytogenetic Analysis, DNA Extraction and WGS

Juvenile individuals of *C. macropomum* (*n* = 6) and *P. mesopotamicus* (*n* = 8) were obtained from the Aquaculture Center of São Paulo State University, Jaboticabal, São Paulo, Brazil. All animals were collected in accordance with Brazilian environmental protection legislation (Collection Permission MMA/IBAMA/SISBIO—number 3245). The sampling and experiments followed the international rules on animal experimentation, followed by the Ethics Committee on the Use of Animals at the São Paulo State University (IBB/UNESP), protocol 1204-CEUA/2019.

Chromosomal preparations with mitotic metaphasic chromosomes were obtained from all the specimens by following the protocol established in [40]. From each specimen, we also fixed liver tissue samples for molecular biology experiments. After analysis, animals were fixed in 10% formaldehyde; preserved in 70% ethanol; and deposited in the fish collection of *Fish Genetics Laboratory*, Faculty of Sciences, São Paulo State University, Bauru, São Paulo, Brazil, under the vouchers LG13046–LG13053 (*P. mesopotamicus*) and LG14119–LG14124 (*C. macropomum*).

Genomic DNA was extracted from liver samples of *C. macropomum* and *P. mesopotamicus* (one individual each), using the Wizard Genomic Kit (Promega, Madison, WI, USA), following the manufacturer’s instructions, with a step of RNA purification using RNAse A (Invitrogen, Waltham, MA, USA). Finally, we checked the DNA quality with 1% agarose gel electrophoresis.

After extraction, genomic DNA samples of *P. mesopotamicus* and *C. macropomum* were sequenced on the Illumina MiSeq (2 × 251 bp) and MGISEQ-2000 (2 × 101 bp) platforms, respectively. Sequencing yielded about 0.49 Gb for *P*. *mesopotamicus* (~0.45× of coverage) and 1.66 Gb for *C. macropomum* (~1.4× of coverage). We deposited the raw read libraries in the sequence read archive of the NCBI under the accession numbers SRR11676685 (*P. mesopotamicus*) and SRR22219024 (*C. macropomum*).

### 2.2. Bioinformatic Analyses—Satellitome Characterization and Comparison

Genomic libraries were initially quality trimmed with Trimmomatic [41]. Then reads of *P. mesopotamicus* and *C. macropomum* were cropped to 101 bp for a standardization of the subsequent analyses. We characterized the satellitomes of both species by performing several iterations of TAREAN tool [42], followed by filtering out the discovered satDNAs with DeconSeq [43], until no satDNA was discovered by TAREAN. In each iteration, we input 2 × 250,000 reads into TAREAN. After that, we identified and removed other tandemly repeated sequences, such as multigene families, that are usually outputted by TAREAN. Then we performed a similarity search by using RepeatMasker software (https://github.com/fjruizruano/satminer/blob/master/rm_homology.py, accessed on 6 September 2022) to detect redundances in the catalog, as well as classify the isolated satDNAs as the same variant of a single satDNA (similarity greater than 95%), variants of a satDNA (similarity between 80 and 95%) and superfamilies (similarity between 50 and 80%), as established by [15].

We estimated the abundance and divergence of each satDNA with RepeatMasker [44], using a custom python script (https://github.com/fjruizruano/ngs-protocols/blob/master/repeat_masker_run_big.py, acessed on 6 September 2022). For this, we randomly selected 2 × 2,457,335 reads in each species (totalizing 0.49 Gb in each species, approximately 0.42× of genome coverage) and mapped them against their own satellitomes, so that the number of mapped reads divided by the number of analyzed nucleotides indicates the relative abundance of each satDNA. After that, satDNA families were named as suggested in [15], with the species abbreviation (Pme for *P. mesopotamicus* and Cma for *C. macropomum*), plus the term “Sat” and the catalog number, in decreasing order of abundance, followed by the size in base pairs of each monomer.

We also performed a comparative analysis between both satellitomes. For this, we performed the following steps: (i) concatenate the catalogs containing consensus sequences of all satDNAs from both species in a single file; (ii) convert the monomer fasta file in a dimer fasta file (https://github.com/fjruizruano/ngs-protocols/blob/master/dimerator.py, accessed on 6 September 2022); (iii) run RepeatMasker to detect similarities among sequences (https://github.com/fjruizruano/ngs-protocols/blob/master/rm_homology.py, accessed on 6 September 2022); and (iv) manually align similar sequences (only monomers) with MUSCLE. Then, if a satDNA present in both species exhibited at least 50% similarity, we considered that they had a common origin. To obtain more information on the conservation and sharing of satDNAs with other species, we also performed the abovementioned analysis encompassing 11 satellitomes of Characiformes [18,22,23,24,25,26,27]. Here, we only considered the consensus sequences with similarity higher than 80%.

### 2.3. Primers Design, Probes Manufacturing and Fluorescence In Situ Hybridization (FISH)

For a cytogenetic approach, we restricted our analyses to a subset of sequences that included the five most abundant satDNAs in each species, as well as all the shared satDNAs between both species (restricted to consensus sequences). In total, we designed primers and generated FISH probes for 20 satDNAs (Appendix A). All of these probes were constructed with the gDNA of the respective species. The probes were labeled with digoxigenin-11-dUTP in PCR reactions. Six conserved satDNAs were not analyzed by FISH due to their monomer sizes being shorter than 40 bp.

FISH was performed as described in [45], with small adaptations, as follows. Chromosomes were treated with 0.005% Pepsin/10 mM HCl for 5 min, fixed in 1% formaldehyde in 1× PBS/50 mM MgCl_2_ for 10 min and dehydrated in ethanol series for 3 min each (70%, 80% and 100%). Then chromosomal DNA was denatured in 70% formamide/2× SSC for 2 min at 70 °C. For each slide, 30 µL of hybridization solution containing 200 ng of the labeled probe, 50% formamide, 2× SSC and 10% dextran sulfate was denatured for 10 min at 95 °C, dropped onto the slides and hybridized overnight at 37 °C in a moist chamber. Post-hybridization, slides were washed in 0.2× SSC/15% formamide at 42 °C, followed by washes in 0.1× SSC for 15 min at 60 °C. Probe detection was performed with anti-digoxigenin-rhodamine, and chromosomes were counterstained with DAPI (4′,6-diamino-2-phenylindole, Vector Laboratories, Burlingame, CA, USA). Images were captured by using an optical microscope (Olympus BX61) with DP Control software (Olympus^®^, Hamburg, Germany). A minimum of 10 cells from each FISH experiment were analyzed to confirm the observed FISH patterns. After FISH, C-banding experiments were performed with the same slides to locate the constitutive heterochromatin, following the protocol established by [46], with minor adaptations.

## 3. Results

### 3.1. Satellitome Description of P. mesopotamicus and C. macropomum Reveals Several Shared satDNAs

After two iterations of TAREAN in each species, we identified 30 and 46 satDNA families in *P. mesopotamicus* and *C. macropomum*, respectively (Table 1 and Table 2). In *P. mesopotamicus*, we found a single superfamily that comprised PmeSat01-508 and PmeSat02-143, the two most abundant satDNAs in this species. The repeat unit lengths (RUL) varied between 6 and 1853 bp, with median a of 286 bp. There was a balance in the number of long satDNAs (>100 bp) and short satDNAs (<100 bp), with 16 and 14 satDNAs represented in these categories, respectively. The A + T content varied between 45.2 and 76.1%, with a median of 59.6%, evidencing some predominance of A + T rich satDNAs.

In *C. macropomum*, we detected four superfamilies, as follows: (1) CmaSat01-144 and CmaSat04-141; (2) CmaSat02-543 and CmaSat08-285; (3) CmaSat05-247 and CmaSat32-237; and (4) CmaSat11-234 and Cma24-327. The RUL varied between 6 and 3068 bp, with a median of 429 bp. Moreover, there was a predominance of long satDNAs families, with 27 sequences showing more than 100 bp. The A + T content revealed the predominance of A + T rich satDNAs, as observed in *P. mesopotamicus*, varying between 43.3 and 76.2% (59.3% median).

The comparative analyses revealed that *P. mesopotamicus* and *C. macropomum* share 21 satDNAs, showing at least 50% of similarity. One of these corresponds to the telomeric sequence that is conserved in vertebrates [47]. Thus, from the remaining twenty satDNAs, five were considered as the same superfamily (SF, similarity between 50 and 80%), six were considered variants of the same satDNA (similarity between 80 and 95%) and eight were considered same variants (similarity greater than 95%). These results are summarized in Table 3, and the corresponding sequence alignments are shown in Appendix A.

The comparative analysis among all the available characiform satellitomes revealed that ten satDNAs present in *C. macropomum* or *P. mesopotamicus* are shared with at least one Characiformes species. Thus, nine satDNAs that were reported as being shared between *C. macropomum* and *P. mesopotamicus* were found in other Characiformes, while CmaSat35-30, which was only found in *C. macropomum*, was also detected in *M. macrocephalus*, *M. elongatus* and *P. lineatus*. These 10 shared satDNAs were all short satDNAs (<100 bp), with a maximum RUL of 76 bp. All of these results are summarized in Table 4 and Appendix A.

### 3.2. FISH Reveals Maintenance of Chromosomal Clustering Sites of satDNAs in P. mesopotamicus and C. macropomum

The analyzed specimens exhibited a diploid chromosome number of 2*n* = 54, with predominance of metacentric and submetacentric chromosomes. C-banding experiments revealed C-positive blocks in all chromosomes of *P. mesopotamicus* and *C. macropomum* in pericentromeric and telomeric chromosomal regions (Figure 1).

We performed the cytogenetic mapping of the five most abundant satDNA families in *P. mesopotamicus* and *C. macropomum* (Figure 1). We observed a general accumulation of satDNAs in pericentromeric and telomeric regions in both species. In *P. mesopotamicus*, satDNAs are clustered in pericentromeric regions, except PmeSat01-508, which is accumulated on the telomeres. Notably, PmeSat05-247 clusters were detected in all centromeres, but not in all the analyzed metaphases, which could be attributed to the existence of small chromosomal clusters (Appendix A). In *C. macropomum*, CmaSat01-144 and CmaSat03-177 are clustered on the telomeric regions of the same chromosomes, while CmaSat05-247 and CmaSat02-543 are located on pericentromeric regions. CmaSat04-141 and PmeSat02-143 exhibited a similar pattern of distribution, with signals on the pericentromeric and telomeric region of one chromosomal pair and two additional pairs of chromosomes in *P. mesopotamicus* (Figure 1).

We also performed FISH with the conserved satDNAs, showing at least 50% similarity (Figure 2). In general, these experiments revealed similar patterns of clustering in both species, with four main different chromosomal locations: (i) pericentromeric (four satDNAs), (ii) telomeric (four satDNAs), (iii) dispersed (one satDNA) and (iv) non-clustered (one satDNA) (Figure 2). The number of clusters per species varied for some satDNAs (e.g., PmeSat02/CmaSat04 and PmeSat11/CmaSat36), while for others (e.g., PmeSat14/CmaSat16 and PmeSat15), the number of clusters was also similar. All the results are summarized in Table 5.

## 4. Discussion

Considering the new data obtained here, 11 satellitomes are now available for Characiformes, and general common features could be identified. Thus, satDNA families are numerous in this group (average of 56 satellites per species) when compared to other vertebrates, such as birds (average of 6.7 satDNAs per species; [5]) and crocodilians (average of 7.8 satDNAs per species; Oliveira in prep.). We reinforce that our sampling is restricted to a single order within Teleostei and that the characterization of satDNA catalogues from other taxa will certainly contribute for a better understanding of satDNA biology in eukaryotes.

SatDNAs are among the fastest evolving sequences in eukaryotes [2]. While highly abundant, especially in heterochromatic areas, our knowledge about these sequences is still limited, mainly because satDNA catalogues are not yet available for a considerable number of species. In Neotropical fishes, satellitomes were characterized in different Characiformes species, encompassing eight species within five families [18,22,23,24,25,26,27]. Here, we added two species from the Serrasalmidae family to this list, providing valuable genetic resources for future research, including the delimitation of satDNAs located on areas that are difficult to assemble, as pericentromeric and telomeric regions.

Previous studies had already shown that C-heterochromatin is not restricted to the commonly observed pericentromeric position in *C*. *macropomum* and *P*. *mesopotamicus*, but also in interstitial and telomeric regions [37,48]. Here, we mapped the five most abundant satDNAs in both species with FISH. As predicted by theory [49], centromeric and non-centromeric C-heterochromatin were highly enriched in satDNAs (Figure 1), and this is probably related to the well-known amplification and spread mechanisms such as unequal crossing-over and rolling-circle amplification [50]. Moreover, we could observe that clusters of the most abundant satDNAs in *C. macropomum* and *P. mesopotamicus* were not present in all the heterochromatic blocks on the telomeres. This fact suggests that telomeric heterochromatic blocks in *C. macropomum* and *P. mesopotamicus* are constituted by other repetitive DNAs. Indeed, the authors of [51] detected an accumulation of Rex3 transposable elements on the heterochromatic telomeric areas of *C. macropomum*, providing evidence that other repeated DNAs constitute these areas. Remarkably, the presence of *Rex3* sites on these areas could facilitate the dispersion and accumulation of other sequences [52].

Centromeres show a conserved function in all eukaryotes and can be defined as the chromosomal domain where the kinetochore is assembled [53]. They are usually composed of long arrays of satDNAs that evolve rapidly, with examples of closely related species that exhibit completely different repeated DNA sequences [54,55]. In some species, a single repeat can be found in all centromeres [14], while others show centromere-specific satellite repeats [56]. In this context, the most abundant satDNA from a given species is usually the centromeric satDNA [57]. While recent studies confirmed this assumption for some Neotropical fishes [25,27], this was not true for other ones [18,24,26]. Here, centromeric satDNAs were the fifth in abundance in both species (CmaSat05 and PmeSat05), while the most abundant satDNAs were located on several telomeres (CmaSat01 and PmeSat01). These results illustrate the dynamics of satDNAs and evidence the importance of physical mapping in cytogenomic studies.

The long-term evolution of satDNAs usually follows the library hypothesis, which proposes that closely related species share a set of satDNAs that was present in a common ancestor, and mainly quantitative changes are detected in closely related species [9]. Our results are in accordance with this model, as *P. mesopotamicus* and *C. macropomum* share several satDNAs, with 15 of them showing at least 80% of similarity in consensus-against-consensus alignments, although they diverged approximately 30 mya. Thus, some of these sequences seem to have originated and amplificated in the common ancestor and remained in the same genomic regions for millions of years. For instance, PmeSat14-956/CmaSat16-955 (91.4% of similarity) is located on the pericentromeric region of a single chromosome pair (No. 3) in both species, while PmeSat08/CmaSat03 (87% of similarity) is more abundant in *C. macropomum*. These results are notable, mainly considering the clustered satDNAs that would be under higher concerted evolution effects than non-clustered satDNAs [10], which would inexorably lead to higher differentiation over time in different species [58]. For these fishes, their long-life cycles (i.e., 4 or 5 years to reproduce) could slow down rates of satDNA differentiation, explaining the conservation of such sequences after millions of years. Importantly, our results do not imply that all the other satDNA families are not shared between both species, since we restricted our analyses to TAREAN-derived sequences, and previous studies have already shown that less abundant tandem repeats are not always clustered with TAREAN [16].

Our comparative analysis considering other Characiformes species revealed nine shared satDNAs that are highly conserved (>80% of similarity). Although this is primarily contrasting with evolutionary patterns of satDNAs, we noted that almost all of them exhibit short RULs and are not highly abundant and clustered on the chromosomes of the analyzed species, such as PmeSat12-72, PmeSat21-54, PmeSat17-65 and PmeSat18-67. Thus, we hypothesize that this extreme conservation of particular satDNAs among distantly related species could be due to the organization as short arrays that would escape from the effects of concerted evolution in large arrays. Interestingly, two conserved satDNAs (i.e., PmeSat12-72/CmaSat20-72/PliSat15-75 and MelSat08-42/CmaSat09-42) were clustered in one of the analyzed species; this could be explained by satDNAs being favored by constraints imposed on the sequence in the heterochromatic environment [12,59,60,61,62,63]. One must note that functional constraints cannot be ruled out in any of these cases, as well.

## Figures and Tables

**Figure 1 genes-14-00091-f001:**
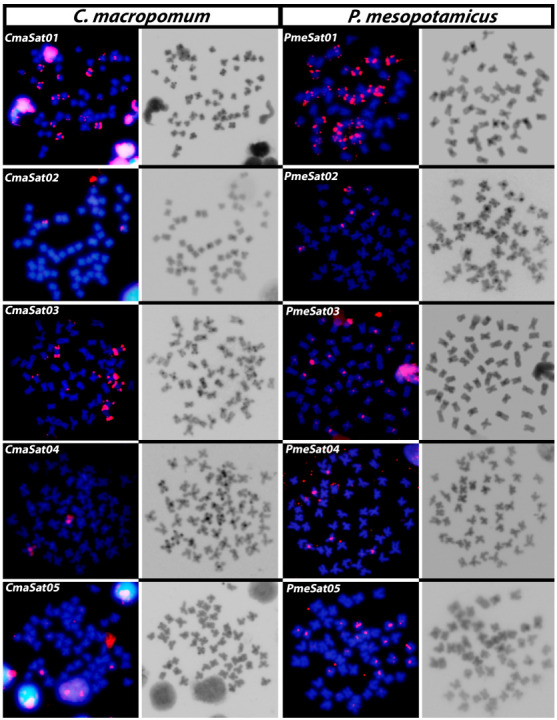
Metaphase plates of *C. macropomum* and *P. mesopotamicus*, highlighting the chromosomal location of the five most abundant satDNAs in each species and the patterns of constitutive heterochromatin.

**Figure 2 genes-14-00091-f002:**
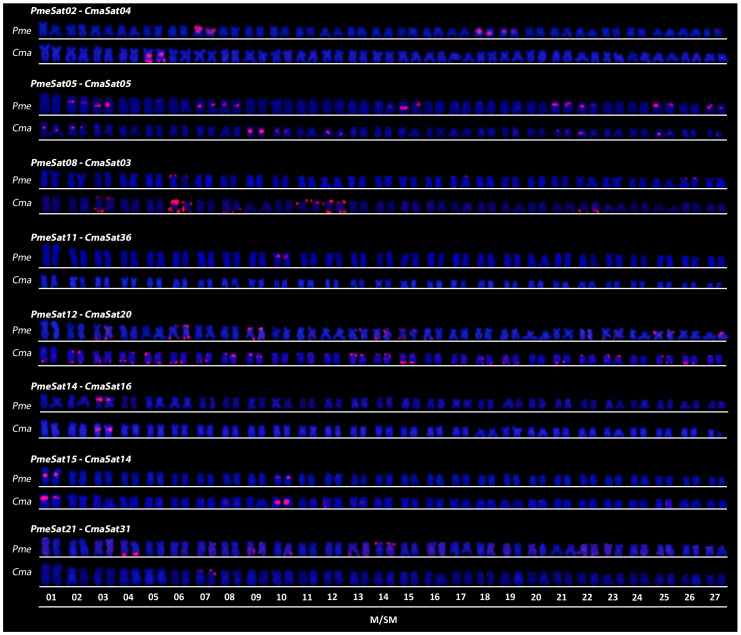
Karyotypes of *P. mesopotamicus* (Pme) and *C. macropomum* (Cma) after FISH with shared satDNAs.

**Table 1 genes-14-00091-t001:** Main features of Piaractus mesopotamicus satellitome.

satDNA	RUL	A + T (%)	Abundance	Divergence (%)
PmeSat01-508	508	58.6	0.00938	8.10
PmeSat02-143	143	55.9	0.00837	10.74
PmeSat03-1068	1068	58.8	0.00752	11.85
PmeSat04-118	118	62.7	0.00322	10.75
PmeSat05-247	247	61.9	0.00192	18.62
PmeSat06-1853	1853	56.1	0.00173	8.38
PmeSat07-42	42	52.3	0.00112	15.06
PmeSat08-177	177	65.5	0.00092	14.96
PmeSat09-696	696	66.6	0.00076	7.88
PmeSat10-21	21	76.1	0.00069	16.97
PmeSat11-1242	1242	58.5	0.00064	1.82
PmeSat12-72	72	68.0	0.00053	6.72
PmeSat13-92	92	73.9	0.00051	4.75
PmeSat14-956	956	63.2	0.00051	1.48
PmeSat15-157	157	50.3	0.00051	5.05
PmeSat16-42	42	45.2	0.00044	12.12
PmeSat17-65	65	63.0	0.00038	6.26
PmeSat18-67	67	65.6	0.00036	9.11
PmeSat19-30	30	56.6	0.00035	10.15
PmeSat20-142	142	57.0	0.00033	13.53
PmeSat21-54	54	51.8	0.00031	10.04
PmeSat22-28	28	50.0	0.00030	10.87
PmeSat23-38	38	63.1	0.00026	8.25
PmeSat24-83	83	54.2	0.00023	9.93
PmeSat25-6	6	50.0	0.00021	23.98
PmeSat26-81	81	64.1	0.00021	7.68
PmeSat27-102	102	59.8	0.00019	11.35
PmeSat28-398	398	57.2	0.00017	6.54
PmeSat29-30	30	60.0	0.00016	13.14
PmeSat30-33	33	63.6	0.00006	5.67

**Table 2 genes-14-00091-t002:** Main features of *C*. *macropomum* satellitome.

satDNA	RUL	A + T (%)	Abundance	Divergence (%)
CmaSat01-144	144	56.2	0.01389	6.62
CmaSat02-543	543	58.0	0.00534	13.07
CmaSat03-177	177	64.9	0.00515	4.27
CmaSat04-141	141	57.4	0.00464	5.22
CmaSat05-247	247	63.1	0.00447	26.04
CmaSat06-3168	3168	57.0	0.00399	14.69
CmaSat07-525	525	57.1	0.00318	1.99
CmaSat08-285	177	58.5	0.00280	8.90
CmaSat09-42	42	54.7	0.00166	12.92
CmaSat10-43	43	48.8	0.00158	19.89
CmaSat11-234	1242	57.2	0.00130	5.46
CmaSat12-1846	1846	66.3	0.00103	5.91
CmaSat13-2220	2220	60.0	0.00100	12.00
CmaSat14-170	170	56.4	0.00095	0.03
CmaSat15-21	21	76.1	0.00089	17.19
CmaSat16-955	955	63.2	0.00076	0.73
CmaSat17-1946	1946	53.5	0.00060	7.25
CmaSat18-30	30	56.6	0.00058	9.28
CmaSat19-38	38	65.7	0.00055	6.71
CmaSat20-72	72	65.2	0.00054	3.10
CmaSat21-28	28	53.5	0.00051	10.39
CmaSat22-68	68	63.2	0.00049	5.11
CmaSat23-898	898	59.9	0.00046	12.74
CmaSat24-327	327	52.5	0.00046	3.34
CmaSat25-6	6	50.0	0.00046	20.53
CmaSat26-50	50	64.0	0.00043	4.40
CmaSat27-394	394	55.5	0.00041	4.28
CmaSat28-183	183	53.5	0.00035	4.23
CmaSat29-34	34	70.5	0.00034	17.93
CmaSat30-919	919	67.2	0.00034	14.39
CmaSat31-54	54	53.7	0.00034	8.78
CmaSat32-237	237	59.4	0.00032	16.73
CmaSat33-66	66	65.1	0.00032	6.47
CmaSat34-101	101	60.3	0.00031	6.93
CmaSat35-30	30	53.3	0.00029	11.94
CmaSat36-1250	1250	59.9	0.00024	2.49
CmaSat37-51	51	62.7	0.00021	2.66
CmaSat38-30	30	63.3	0.00019	11.07
CmaSat39-192	192	61.9	0.00015	7.47
CmaSat40-932	932	51.0	0.00014	2.00
CmaSat41-263	263	57.7	0.00014	9.78
CmaSat42-83	83	60.2	0.00013	10.92
CmaSat43-62	62	61.2	0.00012	7.20
CmaSat44-324	324	68.2	0.00011	9.01
CmaSat45-289	289	62.2	0.00010	4.82
CmaSat46-30	30	43.3	0.00009	6.48

**Table 3 genes-14-00091-t003:** Shared SatDNAs between *C. macropomum* and *P. mesopotamicus*. Highlighted satDNAs were observed in other characiform satellitomes.

*P. mesopotamicus*	*C. macropomum*	Similarity (%)	Classification
PmeSat02-143	CmaSat04-141	78.3	SF
PmeSat02-143	CmaSat01-144	67.5	SF
PmeSat07-42	CmaSat09-42	97.6	SV
PmeSat05-247	CmaSat05-247	72.5	SF
PmeSat05-247	CmaSat32-237	59.7	SF
PmeSat08-177	CmaSat03-177	87	V
PmeSat10-21	CmaSat15-21	100	SV
PmeSat11-1242	CmaSat36-1250	87	V
PmeSat12-72	CmaSat20-72	97.2	SV
PmeSat14-956	CmaSat16-955	91.4	V
PmeSat15-170	CmaSat14-157	73.5	SF
PmeSat17-65	CmaSat22-68	95.5	SV
PmeSat18-67	CmaSat33-66	98.5	SV
PmeSat19-30	CmaSat18-30	100	SV
PmeSat21-54	CmaSat31-54	98.1	SV
PmeSat22-28	CmaSat21-28	96.4	SV
PmeSat23-38	CmaSat19-38	81.5	V
PmeSat25-6	CmaSat25-6	100	SV
PmeSat27-102	CmaSat34-101	97	SV
PmeSat28-398	CmaSat27-398	91.2	V
PmeSat29-30	CmaSat38-30	90	V

**Table 4 genes-14-00091-t004:** Shared SatDNAs between *C. macropomum*, *P. mesopotamicus* and other *Characiformes*. Cma, *Colossoma macropomum*; Pme, *Piaractus mesopotamicus*; Mma, *Megaleporinus macrocephalus*; Mel, *Megaleporinus elongatus*; Cgo, *Characidium gomesi*; Pli, *Prochilodus lineatus*; and Tau, *Triportheus auratus*.

Cma	Pme	Mma	Mel	Cgo	Pli	Tau
CmaSat09-42	PmeSat07-42	MmaSat07-42	MelSat08-42	CgoSat10-42	PliSat12-42	TauSat06-42
CmaSat15-21	PmeSat10-21	-	-	-	PliSat17-21	-
CmaSat18-30	PmeSat19-30	-	MelSat25-30	-	PliSat19-30	-
CmaSat20-72	PmeSat12-72	-	-	-	PliSat15-75	TauSat19-76
CmaSat21-28	PmeSat22-28	MmaSat41-29	MelSat48-29	CgoSat26-29	-	TauSat16-29
CmaSat22-68	PmeSat17-65	MmaSat84-65	MelSat39-65	-	PliSat36-68	TauSat12-66
CmaSat31-54	PmeSat21-54	MmaSat38-54	-	-	-	-
CmaSat33-66	PmeSat18-67	MmaSat27-67	MelSat12-67	-	PliSat16-67	-
CmaSat35-30	-	MmaSat57-30	MelSat58-31	-	PliSat30-31	-
CmaSat38-30	PmeSat29-30	-	MelSat54-30	-	PliSat32-30	-

**Table 5 genes-14-00091-t005:** Fish patterns of SatDNA families conserved between *P. mesopotamicus* and *C. macropomum*. The patterns of satDNAs were classified as clustered (C), non-clustered (NC) and mixed (M). Position was catalogued as pericentromeric (c) and telomeric (t).

*SatDNA*	*C. macropomum*	*P. mesopotamicus*	Position
PmeSat02-143	C	C	c/t
PmeSat05-247	C	C	c
PmeSat08-177	C	C	t
PmeSat11-1242	C	C	t
PmeSat12-72	C	C	t
PmeSat14-956	C	C	c
PmeSat15-170	C	C	c
PmeSat17-65	M	M	-
PmeSat18-67	NC	NC	nc
PmeSat21-54	C	C	t
PmeSat27-102	NC	NC	nc
PmeSat28-398	NC	NC	nc

## Data Availability

The reported results can be found in different databases, and the accession numbers and links were provided in the text: raw reads (sequence read archive, NCBI), satDNA sequences (Genbank, NCBI) and personal bioinformatic scripts (Github).

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
