# Peer review of "The Satellite DNA Catalogues of Two Serrasalmidae (Teleostei, Characiformes): Conservation of General satDNA Features over 30 Million Years"

_genes, 2022, doi:10.3390/genes14010091_

Round 1

Reviewer 1 Report

In the Work entitle “The satellite DNA catalogues of two Serrasalmidae (Teleostei, 2 Characiformes): conservation of general satDNA features over 3 30 million years, the genome of the two analyzed species are sequenced, and satatellite sequences have been analyzed through bioinformatic tools. Furthermore some satellite probes identified through the previous bionformatic tools have been produced to be mapped by FISH on chromosomes.

The idea is very nice and the informatics part is interesting; also Hybridization are good but however the article is just descriptive and very hard to follow. For example when speak about superfamily without never introduce them lines 169-234

Also results are not always true: “general, these experiments revealed similar patterns of clustering in both species” Line 224

In my opinion all the article need to be better address

Reviewer 2 Report

The authors present a paper which characterizes satellitomes of the two Characiformes fish, inspect the levels of satDNA conservation between the analyzed species, in the nucleotide and chromosomal level, using in silico and in situ analyses.

The topic is interesting and it contributes to the research area of repetitive DNA sequences, which is enriched by obtaining information from different model systems and diverse eukaryotic genomes.

However, several things should be attended.
My comments and suggestions are listed below:

Line 36: It would be more precise to state “long arrays of monomers repeated in tandem” instead “long tandemly repeated arrays”, as arrays are not those which are repeated but monomers, which then constitute the arrays.

Line 39: I suggest removing “if any”, as many functions have been proposed for satDNAs. SatDNA contributes to the essential processes of assembly of centromeric chromatin, heterochromatin establishment, dosage compensation, reproductive isolation, genome stability and development. SatDNA sequences are also transcribed and misregulation of their expression is found to lead to various defects in the maintenance of genomic architecture, chromosome segregation and gametogenesis (reviewed in Shatskikh et al. 2020, Front Cell Dev Biol.).

Line 92: “Firstly, cytogenetic analyses were performed for all the specimens.” This seems impossible given the fact that the nucleotide sequence of the probes used for cytogenetic analyses are obtained after the sequencing and bioinformatic analyses. I suppose only slides were prepared at this point, however, it would be good to rephrase this sentence.

Line 116: 2x250,000 reads corresponds to which genome coverage of each species?

Lines 140-142, lines 188-189, line 223:
I am concerned as 50% similarity between 2 nucleotide sequence is quite low and can happen by chance even among 2 not quite related nucleotide sequences. Did the sequences showing such low overall similarity have at least some segments/stretches of higher similarity between them which would support their relatedness more?

C-banding experiments presented in Figure 1 are not mentioned or described in the Materials and methods section.

Lines 147-148: It is not explained weather the probes are labelled by PCR amplification directly from the genomic DNA and contain a mix of sequences belonging to the same satDNA, encompassing different monomer variants, or they are based on one clone with the specific nucleotide sequence.

Line 167-168: I suggest omitting the term “family” and use only “satDNAs”. In several places in the manuscript the following definitions are brought: superfamily (SF – similarity between 50% and 80%), variants of the same satDNA family (similarity between 80% and 95%), same variants (similarity greater than 95%). This is mostly used for inter-satellelite comparisons. However, this raises the question what would be an exact definition of the “satDNA family”? For example, if the same context tries to be applied intra-satellite, some of the satDNAs presented in Tables 1 and 2 named “satDNA family” actually have internal divergence that is greater than 20%, thereby not all sequences of this satDNA meet the criteria proposed for “variants of the same satDNA family”.)
For that reason, to avoid the confusion, please provide the clear definition of the “satDNA family” or please use only “satDNA” instead of “satDNA family” in the Table 1 and 2, lines 167-168, 189, and other related parts of the manuscript.
Additionally, in the continuation of the manuscript both terms “satDNA family” and “satDNA” are interchangeably used, which should be unified (depending on what authors prefer to keep), unless they provide distinct definitions for both terms.

The abundance in Table 1 and 2 is presented with an unnecessarily large number of decimals, I suggest reducing and unifying the number of decimals in the abundance presentation.

Figure 2 and Table 4. What was used as a probe in these FISH experiments? From the Figure 2 it could be concluded that only sequences of PmeSats were used as probes, while the Materials and methods mention “shared satDNAs between both species (restricted to consensus sequences); line 145”. If the probes were consensus sequences for each species, how were they prepared? If they were probes only for PmeSats, was it a mix of monomer variants of this species or one specific clone? If probes for PmeSats were used on Cma, how can we be sure that the FISH results are not incomplete and biased, as some of the shared satDNAs exhibit only 59.7% similarity (e.g. PmeSat05 & CmaSat32), which means that not necessarily all loci/variants are detected in the other species if the probe is specific for PmeSats. Please clarify, and add more detailed explanation in the M&M and the Results sections.

In Table 4, six satDNAs are blank/without data, and in the Material and methods it is stated that “Five conserved satDNAs were not analyzed…”.

Lines 261-364: The authors do not comment on very interesting data they have obtained. In C. macropomum the most abundant satDNAs do not colocalize with all the C-bands on the chromosomes. This would indicate that heterochromatin in this species is mostly constituted by some other sequences, and the inspected most abundant satDNAs contribute to it only partially.

The authors comment on the satDNA library based on the data from the two species belonging to Characiformes. Despite the fact that the authors propose, that the genomes of the two species evolve slowly, to be able to make more solid conclusion regarding the satDNA library model in this group of organisms, other related species should be inspected. They have mentioned that within Characiformes eight species within five families have defined satellitomes. I would suggest to make a brief comparison of the existing satellitomes with the 2 newly-defined to gain stronger conclusion on satDNA evolution in this group of organisms.

Lines 281-286: The authors conclude that the results are not in accordance with the expectations of satDNA evolution by concerted evolution. However, it has to be considered that several other principles govern the evolution of these sequences, and if the data are observed in the light of all the trends involved, they are in concordance with them (SatDNA evolution follows the library model which proposes that closely related species share a set of satDNA families that originated from the common ancestor, with every satDNA being differentially amplified and propagated in each of the species. Another specificity of satDNA evolution is slow accumulation of mutations, and their DNA sequences can remain “frozen” during very long evolutionary periods. This phenomenon is usually explained as a consequence of functional constraints over a particular sequence and/or as a result of the concerted evolution. However, if the sequence is not involved in vital functional interactions, mutations and rearrangements of an ancestral sequence can form species-specific satDNA variants. Thus, depending on sequence dynamics of a particular satDNA sequence in a particular organism or group of organisms, different evolutionary scenarios may occur, ultimately defining the satDNA landscape of that species).

 Minor observations:

Line 23: Potentially “divergence in the satDNA profiles” could be exchanged with “divergence between the satDNA profiles” to accent that it is inter-satellitomes comparison.

Line 27: I suggest adding also which % of C. macropomum satellitome

Abbreviation “satDNA” has been introduced in line 20, but the full name “satellite DNA” continues to be used in lines 43, 47, 53, 72, etc. Related to that, the title of the manuscript itself contains both the full term “satellite DNA” and its abbreviated form “satDNA”, I don’t know if this should/could be unified.

 Figure S1: The images with the overlap (DAPI+satDNA, upper rows) and the corresponding red counterpart (satDNA signal only) do not seem to have always the same frame. For example, first images for PmeSat05, when comparing the position of red signals in the overlap image and in red only, you can notice that red-only is more zoomed in, and the chromosome visible in DAPI in the right part of the image is not even included in the red-only one. The same is in the first set of images for CmaSat05, red-only is again more zoomed in than DAPI+satDNA, completely omitting some parts of the image (nuclei in the upper part of the image and the chromosome in the left upper part).

Round 2

Reviewer 1 Report

just one point Line 321 I do not see any prrof on the dynamics. Adjust  "dynamics" with different accumulation and distribution

minor poitnts 265/66 name species italics

258 add others references; as it occurs in other taxa as or example:

(Dumas F, Perelman PL, Biltueva L, Roelke ME. 2022. Retrotransposon mapping in spider monkey genomes of the family Atelidae (Platyrrhini, Primates) shows a high level of LINE-1 amplification. J Biol Res doi: 10.4081/jbr.2022.10725

Milioto V, Perelman P L, Paglia LL, Biltueva L, Roelke M, Dumas, F. 2022. Mapping Retrotransposon LINE-1 Sequences into Two Cebidae Species and Homo sapiens Genomes and a Short Review on Primates. Genes, 13(10), 1742.

Reviewer 2 Report

The authors have adequately attended all comments and suggestions.
Therefore, I recommend the manuscript to be accepted for publication in Genes. Congratulations to the authors!